# Laser-induced nitrogen fixation

Huize Wang [1], Ranga Rohit Seemakurthi[2], Gao-Feng Chen [1] ✉,
Volker Strauss [1], Oleksandr Savateev [1], Guangtong Hai[3], Liangxin Ding[4],
Núria López [2] ✉, Haihui Wang [3] ✉ & Markus Antonietti [1]

For decarbonization of ammonia production in industry, alternative methods by exploiting renewable energy sources have recently been explored. Nonetheless, they still lack yield and efficiency to be industrially relevant. Here, we demonstrate an advanced approach of nitrogen fixation to synthesize ammonia at ambient conditions via laser-induced multiphoton dissociation of lithium oxide. Lithium oxide is dissociated under non-equilibrium multiphoton absorption and high temperatures under focused infrared light, and the generated zero-valent metal spontaneously fixes nitrogen and forms a lithium nitride, which upon subsequent hydrolysis generates ammonia. The highest ammonia yield rate of 30.9 micromoles per second per square centimeter is achieved at 25 °C and 1.0 bar nitrogen. This is two orders of magnitude higher than state-of-the-art ammonia synthesis at ambient conditions. The focused infrared light here is produced by a commercial simple $CO_2$ laser, serving as a demonstration of potentially solar pumped lasers for nitrogen fixation and other high excitation chemistry. We anticipate such laser-involved technology will bring unprecedented opportunities to realize not only local ammonia production but also other new chemistries .

The fixation of molecular nitrogen gas ($N_2$) is of great importance for the industrial production of nitrogen-containing fertilizers but also for natural nitrogen chemistry in general[1]. The current method depends upon the Haber-Bosch (H-B) process, which operates at elevated temperatures (400-500 °C) and pressures (100-200 bar). It consumes 1-2% of global annual energy and utilizes methane-derived hydrogen, which results in the emission of more than 300 million tons of $CO_2$ per year[2,3] (about 1.5% of the total). In the near future, renewable electricity, such as solar and wind, but especially solar radiation and heat may become cheap alternatives. However, energy-intensive H-B plants requiring continuous operation with an uninterrupted energy supply cannot handle the intermittency and decentralized character of renewable energy sources[4].

Recently, several techniques have emerged which are compatible with the intermittent nature of renewable energy, that are chemical looping, electrochemistry, plasma- and photocatalysis, and mechanochemical methods[5,6]. Some reports have shown appealing performance improvements, but all these methods still lack yield and rate to be practically relevant, as displayed in a summary of the key results reported (Supplementary Table 1)[2,7–13]. For example, an electrochemical approach working via lithium-mediated $N_2$ fixation, was described. An unprecedentedly high yield rate of 150 nmol s$^{-1}$ cm$^{-2}$ at room temperature and 15-bar $N_2$ was achieved, still lower than commonly defined minimum for practical application, 900 nmol s$^{-1}$ cm$^{-2}$ (obtained at a current density of 300 mA cm$^{-2}$ and a faradaic efficiency of 90%)[14–16]. A further increase of the yield rate is challenging because of the kinetic barrier of ionic diffusion from the bulk electrolyte to the electrode interface and the promotion of side reactions (i.e., decomposition of electrolyte in organic system) at the higher overpotentials needed for higher rates[14,17].

To address the above challenges, we analyze a laser-pulse driven chemical conversion system operating under 1.0 bar $N_2$ pressure[18,19], which enables a one-step, solvent-free transformation of metal oxide to metal nitride, which subsequently can be hydrolyzed to generate

[1]Department of Colloid Chemistry, Max Planck Institute of Colloids and Interfaces, Research Campus Golm, Potsdam, Germany. [2]Institute of Chemical Research of Catalonia (ICIQ-CERCA), The Barcelona Institute of Science and Technology (BIST), Tarragona, Spain. [3]Beijing Key Laboratory for Membrane Materials and Engineering, Department of Chemical Engineering, Tsinghua University, Beijing, China. [4]School of Chemistry and Chemical Engineering, South China University of Technology, Guangzhou, China. ✉e-mail: gaofeng.chen@mpikg.mpg.de; nlopez@iciq.es; cehhwang@tsinghua.edu.cn

ammonia. In our approach, three advantages are met in the laser induction reaction system: (i) laser-induction technology satisfies the demand of small-scale, distributed production of chemicals with the energy converted from broad-band solar radiation;[20,21] (ii) an important characteristic of the focused light is the energy highly concentrated in certain space, which allows non-linear, non-equilibrium chemistry with a potentially high rate for ammonia production; and (iii) different from the electrochemical method and photocatalysis, in which interfacial transport and diffusion of substances, solvents stability, etc. should be taken into account, our system involves only a high photons flux which interacts directly with the bulk phase of high-density metal oxide powder. Accordingly, multiple issues, such as ionic diffusion limits and competing side reactions, are effectively minimized, while the reaction can be carried out selectively and efficiently at a high yield rate.

## Results

### Products characterization of laser-induced N$_2$ fixation

As illustrated schematically in Fig. 1, the laser-induced N$_2$ fixation (LINF) is presented, where a focused-light-induced lithium cycle is based on a Li$_3$N intermediate. Specifically, lithium oxide powder loaded on a titanium sheet substrate is placed in a custom-made reactor (Supplementary Fig. 1). The system is filled with 1.0–7.5 bar nitrogen. The pulsed CO$_2$ laser beam passes through the ZnSe window and focuses on the powder, inducing multiphoton heating and thermal dissociation of Li$_2$O. The generated active metal instantaneously reacts with the N$_2$ atmosphere to form Li$_3$N, as proven by X-ray diffraction (XRD) patterns (Fig. 2a), X-ray photoelectron spectroscopy (XPS) spectra (Fig. 2b)[22,23], Scanning electron microscopy (SEM, Fig. 2c), and energy-dispersive X-ray spectroscopy (EDS, Fig. 2d). Li$_3$N is subsequently hydrolyzed into NH$_3$. The liberated LiOH can be reused in the laser-induced cycle after thermal dehydration.

### Performances Evaluation of ammonia production

The amount of ammonia produced through the hydrolysis of the Li$_3$N is calculated as NH$_3$ yield rate normalized to the loading geometric area of oxides film on Ti sheet substrate (Supplementary Fig. 2). A very high yield rate of 43.3 $\mu$mol s$^{-1}$ cm$^{-2}$ was obtained with 7.5 bar nitrogen when using a light power of 118 kW cm$^{-2}$ (26.9 W) and a scanning speed of 1.36 mm s$^{-1}$ (Fig. 3a, Supplementary Fig. 3). This is equivalent to a

current density of 12.5 A cm$^{-2}$ in the electrochemical method at a hypothetical 100% efficiency. The obtained NH$_3$ yield rate is two orders of magnitude higher than the commonly recognized minimum (300 mA cm$^{-2}$) for practical application of the electrochemical method[14]. The corresponding lowest energy consumption of ammonia synthesis based on the light power can be calculated to be approximately 322 kWh kg$^{-1}$ NH$_3$ (Fig. 3b). This value is significantly higher than that (10 to 13 kWh kg$^{-1}$ NH$_3$) of the H-B process at an industrial scale, but is already competitive in comparison with the H-B process at a laboratory scale (400 °C, 1.0 bar) and especially with the other new emerging methods (Fig. 3c, Supplementary Table 1)[2,5,7,11–13,24–26]. A more favorable economy using scaled processes with heat management and solar light pumping can however be expected. Such non-linear processes as well as the high reaction enthalpies of Li$_3$N hydrolysis create a lot of heat, which then has to be used differently, as in many focused solar light applications.

An experiment using 1.0-bar isotope-labeled $^{15}$N$_2$ gas unambiguously proves that ammonia originates from nitrides obtained by the LINF process. Here, we use nuclear magnetic resonance (NMR) spectroscopy to discriminate $^{15}$NH$_3$ from $^{14}$NH$_3$ and quantify the amount of $^{15}$NH$_3$ produced (See details in methods, Supplementary Figs. 4–8)[11]. As shown in Fig. 3d, a doublet with $J(^{15}\text{N}-^1\text{H}) = 73.6$ Hz is observed in $^1$H NMR spectrum, which is assigned to $^{15}$NH$_3$ without contamination with $^{14}$NH$_3$. Upon integration, a $^{15}$NH$_3$ yield rate of 32.8 $\mu$mol s$^{-1}$ cm$^{-2}$ is calculated. The obtained $^{15}$NH$_3$ yield rate is close to that (30.9 $\mu$mol s$^{-1}$ cm$^{-2}$) of $^{14}$NH$_3$ yield rate obtained from the above experiment using 1.0-bar $^{14}$N$_2$ gas as feedstock (Supplementary Figs. 9–10). Other control experiments using argon with laser treatment and $^{14}$N$_2$ without laser treatment show no NH$_3$ generated.

### Mechanism study of the LINF process

From the above results, it is clear that the LINF converts oxides into nitrides under nitrogen atmosphere. However, it is unclear whether (1) the oxide is converted into nitride in one step, or (2) the metal-oxide bond thermally dissociates under the formation of metal, which then reacts spontaneously with nitrogen and forms metal nitride. Our experimental results suggest that the latter pathway is implemented. Although zero-valent lithium is not detected, zero-valent Mg, Al, Ca, and Zn are left to be detected when using other metal oxides (MgO, Al$_2$O$_3$, CaO, and ZnO) due to their lower reactivity

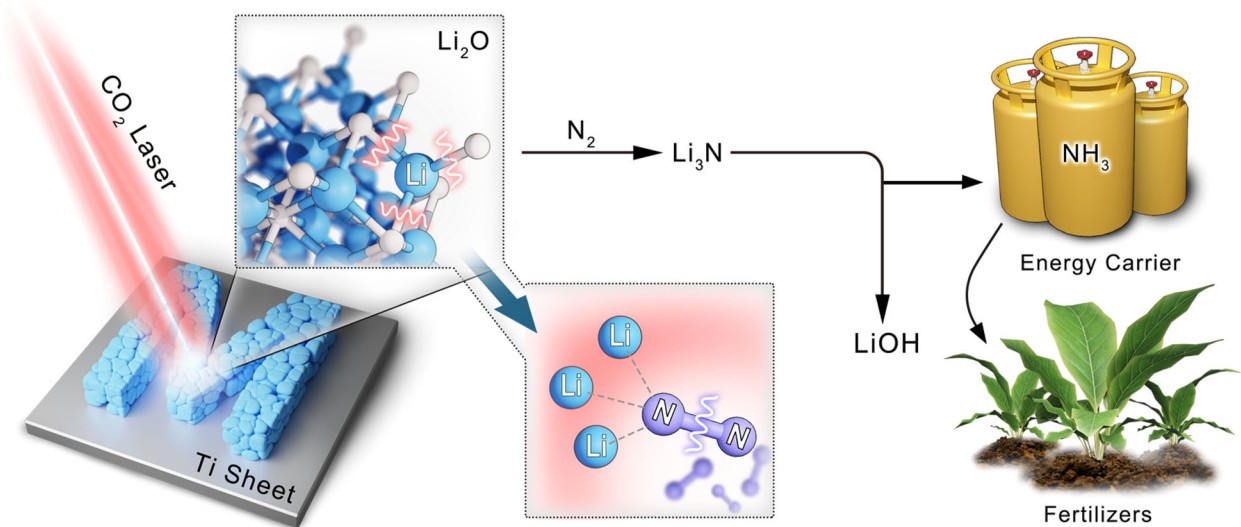

**Fig. 1 | Schematic illustration of the laser-induced nitrogen fixation process.** Lithium oxide powder is placed on the titanium sheet, and by laser-induction, the lithium-oxygen bond is dissociated to form metallic lithium. The activated lithium reacts with nitrogen to form lithium nitride. Lithium nitride is hydrogenated into NH$_3$ as an energy carrier and raw materials for fertilizer production. The lithium hydroxide obtained by hydrolysis can be directly used in laser-induced cycling.

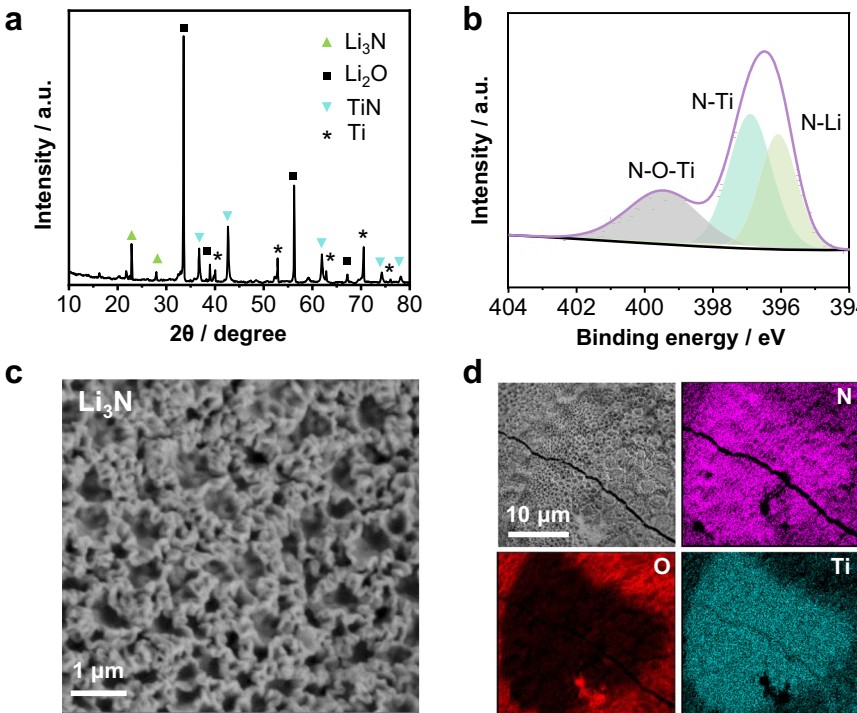

**Fig. 2 | Characterization of metal nitrides. a** XRD patterns of laser-induced Li₃N with the reference XRD patterns of: Li₃N (01-075-8959), Li₂O (01-086-3380), TiN (98-000-0339), Ti (04-003-5042); **b** N₁ₛ region of *XPS* spectra of laser-induced Li₃N film: A strong N signal is observed on the surface and the N₁ₛ region displays three peaks at 396.1, 396.9, and 399.5 eV, which are assigned to N-Li[22], N-Ti and N-O-Ti[23], respectively; **c** Top-view SEM micrograph of Li₃N film with **d** correlated EDXS element maps. It is worth noting that in the LINF process, Ti (substrate) also reacts with N₂ to form TiN following a thermal nitridation process (Supplementary Fig. 28–29, Supplementary Table 3).

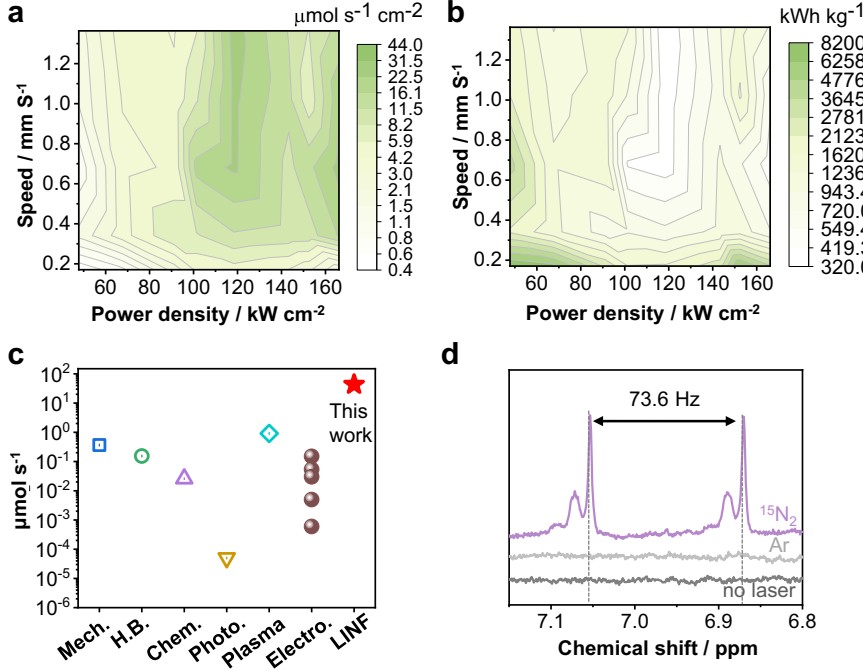

**Fig. 3 | NH₃ production from laser-induced nitrogen fixation. a** 2D plot of ammonia yield rate by using Li₂O as a precursor vs. laser power density and scanning speed at 7.5 bar; **b** 2D plot of the corresponding energy consumption by using Li₂O as a precursor vs. laser power density and scanning speed at 7.5 bar; **c** Comparison of the maximum ammonia yield of this work with other ammonia synthesis methods: Mechanochemical[13], Haber-Bosch[24], chemical looping[25], photochemical[26], plasma electrolytic[5], and electrochemical;[2, 7,9–11] **d** The NMR data from experiments by using ¹⁵N₂ and Argon with laser treatment and ¹⁴N₂ without laser treatment.

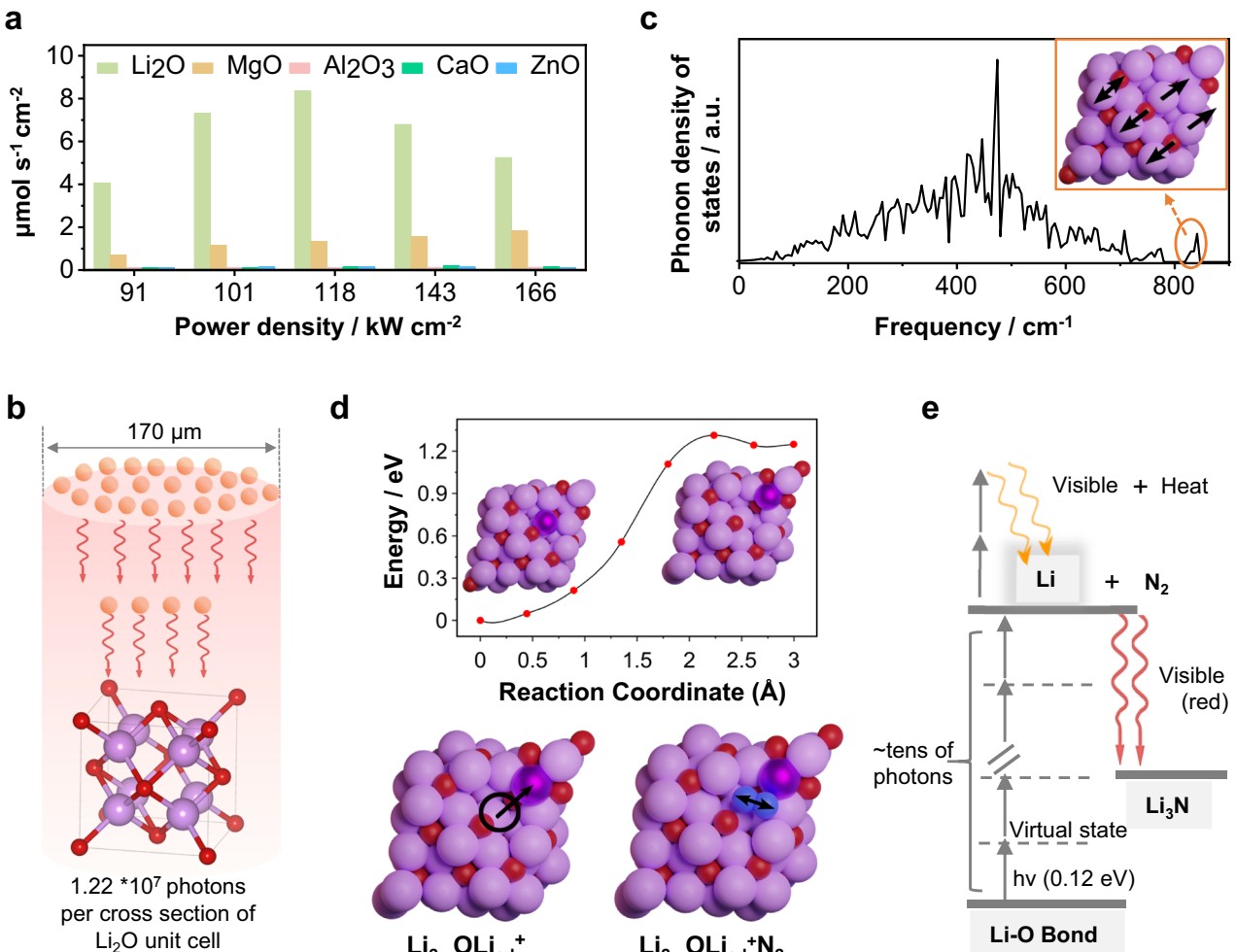

**Fig. 4 | Proposed mechanism of laser-induced nitrogen fixation. a** Ammonia yield rate by using lithium oxide, magnesium oxide, aluminum oxide, calcium oxide and zinc oxide as precursor mediators: a scanning speed of 0.17 mm s⁻¹ and an N₂ pressure at 7.5 bar; **b** The number of photons gathered on a single Li₂O unit cell (purple ball represents lithium atom; red ball represents oxygen atom) by using laser power of 118 kW cm⁻² with a pulse (75 μs) and a laser focus diameter of 170 μm; **c** The phonon density of states of Li₂O (211) surface, the inset of the figure shows the atom movements corresponding to high frequency mode (843.7 cm⁻¹); **d** Nudged elastic band calculation for the generation of the adatom (Li$_{ad}$) defect in the Li₂O

(211) surface. N₂ adsorption in the vacancy close to the adatom with N-N bond distance stretching to 1.165 Å (gas phase N-N = 1.115 Å). Li adatom is shown in the dark purple color, lithium atoms as purple, oxygen atoms as red, nitrogen atoms as blue. **e** Schematic illustration of multiphoton absorption by Li₂O as an example during the laser-induced process. After each oxygen-lithium bond absorbs at least tens of photons (the energy of each photon is 0.12 eV), the oxygen-lithium bond is dissociated, and the excited state lithium transitions to a lower energy level and emits bright light. Part of the excited state lithium is combined with nitrogen gas forms lithium nitride and emits visible red light.

towards N₂ compared to lithium metal (Supplementary Figs. 11–24, Supplementary Note 1), which agree well with the fact that using Li₂O as a medium produces NH₃ with the highest rate (Fig. 4a, Supplementary Fig. 25). Thereby, considering that the required bond dissociation energy of Li₂O (341 kJ mol⁻¹) is lower than that of MgO (394 kJ mol⁻¹), Al₂O₃ (512 kJ mol⁻¹), and CaO (464 kJ mol⁻¹) (Supplementary Table 2), it is reasonable to infer that Li₂O also goes through the dissociation process to form zero–valent lithium first.

Overall, the experimental results support that the reaction mechanism is based on the activation of Li₂O followed by the reaction of a zero–valent metal with N₂. The energy of a single IR photon (170 μm, 0.12 eV) employed in LINF is not sufficient to break Li–O bond (BDE = 3.5 eV). However, the non–linear absorption of about tens of photons is required for the promotion to zero-valent Li when the microseconds pulse with a power density greater than 10⁷ W cm⁻² is applied[27,28]. Bloembergen et al. conjectured that multiphoton dissociation might also occur at power flux as low as ~10³ W cm⁻² when energy is provided within second–long pulses[28]. In our PMNF, by using a laser power density of $1.18 \times 10^5$ W cm⁻² with a pulse duration of 75

microseconds (a frequency of 1000 Hz), we achieve an energy flux of 118 J cm⁻² (Fig. 4b), which is, for example, 64 times greater than the threshold level of 1.84 J cm⁻² required to reach a regime of multiphoton absorption – in this regime the SF₆ is dissociated by absorption of about 30 infrared photons[27]. Our current data can be recalculated to the use of some thousand IR-photons per generated lithium, reflecting the still low efficiency on the scale of atoms.

Furthermore, we give a simplified description of the mechanisms in the PMNF process. The multiphoton dissociation of metal oxides can be described by the equation:

$$MO + nPhotons \rightarrow MO^* \rightarrow M + O + heat + light \qquad (1)$$

where MO is a metal oxide bond, $n$ is the absorbed number of photons, MO* is the excited metal oxide bond unit, M and O are the metal and oxygen atoms generated upon dissociation of the bond unit. IR absorption spectrum of Li₂O features a symmetric LiO stretching vibration ($v$1), LiOLi bending ($v$2), and asymmetric LiO stretching ($v$3)[29,30]. Only the $v$3 vibration mode, which is observed as a broad peak

with the wavenumber ranging from 930 to 1066 cm⁻¹ (Fig. S30), can resonate with the 10.6 μm $CO_2$ laser (943 cm⁻¹). Therefore, in PMNF, $Li_2O$ is able to absorb a large number of infrared photons. At a high energy flux density, the $\nu3$ vibration mode is coherently excited, and due to the coupling of the vibration modes, the excited $\nu3$ vibration is equivalent to a periodic external force oscillation, which activates other vibration modes. It remains unclear if molecular dissociation is thermal or indeed by nonlinear saturation of the vibration mode by significantly more than 30 photons, and the truths presumably are in parts based on both pathways, i.e., temperature lowers bond stability and thereby supports spectral dissociation. Density functional theory (DFT) simulations on (111) and (211) facets of $Li_2O$ provide more hints on its potential activation mechanisms in the $CO_2$ infrared laser frequency ranges. (Fig. 4c, d, Supplementary Figure 26–29, supplementary note 2). Specifically, the $Li_2O$ phonon density of states of stepped (211) surface (Fig. 4c) indicates that high-frequency phonon modes (843.7 cm⁻¹) drive towards structural defects: activation of lithium or oxygen atoms or the exchange of lithium and oxygen on the surface (Fig. 4c inset). Consequently, three distinct defects were examined: Li activation as an adatom on the surface, O atom activation as a peroxide, and anti-site O-Li exchange (Supplementary Figure 26–27). Among the investigated defects, Li activation as an adatom showed the most favorable energetics and can lead to a metastable 0-valent species (Supplementary Table 4–6). Here, Li⁺ first migrates to the surface as an adatom with a reaction energy of $\Delta E = 1.25$ eV and an activation energy $\Delta E_{act} = 1.31$ eV (Fig. 4d) following the 843.7 cm⁻¹ phonon normal mode. This leaves a vacancy in the lattice where $N_2$ can subsequently adsorb exothermically ($\Delta E = -0.88$ eV) and get activated from a gas-phase equilibrium distance of 1.115 Å to 1.165 Å (Fig. 4d). Additionally, the investigation of the other two defects also resulted in stretched $N_2$ bonds of 1.270 Å (Supplementary Table 8). This can be considered as the initial step in the $Li_3N$ formation process.

A model of multiphoton absorption for metal-oxygen bond cleavage is shown in Fig. 4e. Specifically, the ground state is excited into virtual continuum states by absorbing over tens of photons to reach the M–O bond dissociation threshold energy level[31]. When excited molecules dissociate into corresponding atoms, the excess energy as well as the energy of rebinding the fragment atoms is released as heat, as well as the emission of photons in the visible range. Special luminescence is indeed noted during the LINF process. Under argon, the emission of white light is observed (Supplementary Movie 2). Under $N_2$, employing $Li_2O$ emission of red light is observed (Supplementary Movie 1), while the emission of green light is observed in the case of using MgO as a medium (Supplementary Movie 3). The heat generated in vibration relaxation is utilized in the cases when MgO, $Al_2O_3$, CaO, and ZnO are used as media, where nitridation of Mg, Al, Ca, and Zn requires temperatures above 700 °C. This also explains why the yield is lower when using these metal oxides compared to lithium (Fig. 4a), which readily reacts with nitrogen to form lithium nitride even at room temperature[17]. In addition, to compare the performance of MgO to $Li_2O$, the formation of the adatom and peroxide defects together with the $N_2$ adsorption on the MgO(100) surface was calculated (Supplementary Table 9–10). The total reaction energy for this pathway on MgO(100) was found to be at least 1.63 eV higher than on the $Li_2O$(111) surface, indicating the lower reactivity of the MgO.

## Discussion

The LINF achieves an ammonia yield rate that is two orders of magnitude higher than in the other emerging synthesis processes and has a substantial competitive advantage in energy consumption. Remarkably, in this lithium recycling approach, the lithium hydroxide obtained through the hydrolysis of lithium nitride can be stimulated by laser at room temperature, yielding ammonia similarly. Additionally, a considerably high yield of 40.3 μmol s⁻¹ cm⁻² of ammonia was achieved under 200 °C (Supplementary Figs. 31–32). Lastly, successful lithium

cycling the $Li_2O \rightarrow Li_3N \rightleftarrows LiOH$ was accomplished, further validating the effectiveness of the process. Furthermore, scaling up the LING process is both straightforward and feasible, with ammonia production of 1.3 mg achievable after only 78 seconds of irradiation (Supplementary Fig. 33). Given the ongoing optimization of laser technology and equipment, the LINF method shows great potential for practical application in industrial production. Under the best conditions obtained so far, the photon utilization efficiency from lithium oxide to lithium nitride is 5% (Supplementary Fig. 34). Adjustment of the substrate oxide film or optimizing the synthesis equipment to further optimize thermal management certainly allows for improving the utilization rate of photons and the conversion efficiency in the reaction. For instance, a large proportion of photons is certainly lost through light reflection due to unmatched film densities of the oxides or is dissipated as heat by pre-nonlinear relaxation in current experiments. In terms of the economic benefits, achieving a photon utilization rate of 40% or higher can yield cost advantages surpassing current industrial ammonia production in the market (Supplementary Fig. 35). Also the final ammonia hydrolysis generates substantial amounts of heat, which requires secondary uses of heat management.

Therefore, we believe that an optimized LINF process using solar light pumping may be a promising approach for low-cost, fast, decentralized, renewable $NH_3$ synthesis. Here the focused light was generated by the technical industry laser, which however only serves as a lab model for a hypothetical solar–light pumped laser converter[20,21]. For the arguments of this article, such solar–pumped laser systems are thereby comparable to currently well-known systems in which solar cells convert solar energy into electrical energy, which is then used with another efficiency factor for electrochemical synthesis. Therefore, unlike the traditional H–B process for ammonia synthesis, which requires a large-scale centralized reactor and capital, shipping, and storage costs, in LINF we apply overall mild reaction conditions and minimize reaction set–up costs, with the non–linear chemistry and high temperatures only occurring for short times in the focus of laser pulses.

## Methods

### Preparation of metal oxide film

$Li_2O$, MgO, $Al_2O_3$, CaO, or ZnO powders (-3.0 mg) are filled into a circular hole with a diameter of 2.1 ± 0.1 cm and a thickness of 0.1 mm in a titanium sheet. Then the powder is dehydrated under argon at 350 °C.

### Laser-induced process

A high-precision laser engraver setup (Speedy 100, Trotec) was equipped with a center wavelength of 10.6 ± 0.03 μm (0.12 eV) $CO_2$ laser for laser-induced nitrogen fixation. Focusing was achieved with a 2.5-inch focus lens providing a focal depth of ≈3 mm and a focus diameter of d = 170 μm. The scanning speed v, generically given in %, was converted into mm s⁻¹. The effective output power P in watts of the laser was measured with a Solo 2 (Gentec Electro-Optics) power meter. The pulsed mode with a frequency of 1000 Hz is set for all laser induction experiments. The resulting photon density per area and each pulse is given by

$$f = \frac{p}{\omega e \pi r^2} \tag{2}$$

where $p$ is effective power in W, $\omega$ is the laser frequency, $e$ is the photon energy, and $r$ is the focus diameter of the laser lens.

The setting laser parameters: laser power between 10.8 and 37.3 W and the laser scanning speed between 0.17 and 1.36 mm s⁻¹ (corresponding reaction times of 53–7 s to scan the reaction area in this work (Table S14)), adjusted for the experiments to improve the ammonia yield rate. A standard laser-induced pattern is two parallel circles

where the larger is 2 cm in diameter and the smaller is 1 cm in diameter. Then the dehydrated oxide film was placed in a closed reaction atmosphere chamber designed to generate an $N_2$ gas pressure (1–7.5 bar). The top of the reactor is equipped with a ZnSe lens that can allow the IR light to go through and focuses on the primary metal oxide film. Before laser treatment, $N_2$ gas (99.999%) flows through the chamber at a rate of 30 mL min$^{-1}$ for 10 min to ensure the $N_2$ atmosphere within the reactor.

Control experiments were carried out using pure argon with laser treatment or using $N_2$ without laser treatment. In addition, for the experiment operating at ambient conditions, feeding $N_2$ gas passed through four solutions before entering the reactor to purify and prepare the gas. First, the gases were passed through 0.1 M NaOH to capture any $NO_x$, then 0.1 M HCl to capture any $NH_3$, and finally through two bottles with polycarbonate (PC) and molecular sieves to capture water in the gaseous stream. For the isotope labeling experiment, 30 mL min$^{-1}$ of argon gas was initially passed through the chamber to remove air. Then 10 mL min$^{-1}$ of $^{15}N_2$ after pre-purification (Sigma-Aldrich, 98 at% $^{15}$N, 5 L) was fed to the reactor.

The choice of titanium sheets as a substrate is primarily attributed to their excellent stability and high melting point. Furthermore, the titanium nitride produced as a result of the LINF process exhibits enhanced stability, thereby augmenting the substrate's overall stability. This desirable characteristic allows for the repeated utilization of the substrate, ensuring its prolonged functionality.

## Data availability

The data supporting the key findings of this study are available within the article and the Supplementary Information. Additionally, DFT datasets have been uploaded to iochem-BD database[32] under accession code: https://doi.org/10.19061/iochem-bd-1-283[33]. Source data are provided with this paper.

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

## Acknowledgements

We express their gratitude for the financial support by Max Planck Society. G.F.C. thanks the Alexander von Humboldt Foundation for a postdoctoral fellowship. H.H.W. thanks the funding supports by the National Key R&D Program of China (Grant No. 2022YFB4002602), Natural Science Foundation of China (22138005) and X'plorer Prize.

R.R.S. would like to acknowledge funding from the European Union's Horizon 2020 research and innovation programme under the Marie Skłodowska-Curie grant agreement no. 754510. N.L. and R.R.S. thank the Spanish Ministry of Science and Innovation (PID2021-122516OB-I00) and Severo Ochoa (CEX2019-000925-S) for the financial support. The Barcelona Supercomputing Center (BSC-RES) is further acknowledged for providing generous computational resources and technical support.

## Author contributions

H.Z.W. and G.F.C. together conducted most of the experiments. G.F.C. proposed the idea. G.F.C., N.L., H.H.W., and M.A. supervised the project. H.Z.W. and G.F.C. analyzed the experimental data and explained the results. R.R.S. and N.L. carried out theoretical simulations and analyzed the data. V.S., O.S., G.T.H., and L.X.D. contributed to discussions. All the authors participated in discussions and the writing of the manuscript;.

## Funding

## Competing interests

The authors declare no competing interests.
