## [Peer Review File · Nature Communications]

Laser-Induced Nitrogen FixationREVIEWER COMMENTS

Reviewer #1 (Remarks to the Author):

This work by Wang et al. demonstrates ammonia production cycle by employing CO₂ laser with lithium oxide on Ti sheet. The high energy laser induces metal nitride from oxide in N₂ atmosphere, generating ammonia. Ammonia synthesis using a laser has been reported before and the yield rate is high enough. Thus, the idea in this work is a new challenge and creates another field for ammonia synthesis. The reviewer has no reservation for its publication after addressing following points:

1. Ammonia yield rate is high, but it appears to measure over a few microseconds. Is it possible for scale up? Although the laser has high energy, it seems to be unsuitable for scale up because laser is concentrated in a certain spot. Also, in terms of stability, all the chemicals are degraded if exposed to the laser for a long time.
2. If Li₂O acts as a catalyst, it must return to original chemical composition, whatever the chemical intermediates are formed. The authors proposed thermal dehydration, but did it consider when calculating energy consumption?
3. There is a lot of information about yield rate, but no information about yield. The conversion of Li₂O is 5%, but how about the yield? Is the yield comparable to other synthesis methods? For readers' better understanding, comparison table with precedent works (methods, such as the thermochemical Haber-Bosch process, electrochemical process, mechanochemical process, and so on).

Reviewer #2 (Remarks to the Author):

In manuscript NCOMMS-23-08213, Wang and coworkers describe an alternative cyclic ammonia synthesis strategy that uses focused light from a laser to convert Li₂O into a reactive lithium surface that can dissociate the N₂ bond and form a Li₃N intermediate. It is well known that Li₃N can then be easily hydrolyzed to NH₃ and LiOH, and Li₂O can be regenerated via subsequent thermal dehydration. The approach described here is reminiscent of other lithium cycling strategies such as that reported by McEnaney, et al (ref 17), with the primary new idea being the use of laser light rather than electricity to reduce oxidized forms of lithium into lithium nitride.

This work is interesting and potentially suitable for this venue, but I have a couple of concerns that must be addressed before I can endorse publication:

1) In Figure 3c, the authors show that one of the main benefits of their laser-induced nitrogen fixation approach is the high rate of ammonia production compared to various alternatives including the lithium-mediated electrocatalytic approach (e.g. refs 9-11). While this is true for most of the electrocatalytic strategies in the literature, in my view authors should actually be comparing their work against the cycling strategy reported by McEnaney, et al (ref 17), which in my view is the closest technology and would be the direct competitor of the one proposed here. In other words, how does the rate of $\text{Li}_2\text{O} \rightarrow \text{Li (s)} \rightarrow \text{Li}_3\text{N}$ via laser-induced activation compare to the rate of lithium ions electroplating and reacting with N_2 at -3V ($\text{Li}^+ \rightarrow \text{Li (s)} \rightarrow \text{Li}_3\text{N}$)? The subsequent hydrolysis of Li_3N will be the same (fast) in both approaches.

2) Since the primary novelty of this work is the laser-induced activation of Li_2O , I believe that there needs to be a more thorough discussion of the techno-economic benefits of using a laser compared to electricity/electrochemistry. The authors suggest that this technology will be economically relevant, but this statement needs more support.

3) The proposed mechanism and associated DFT modeling in Figure 4 are very limited and do not add much beyond what is already known. The relative thermodynamics of Li_2O , Li_3N , and Li can easily be calculated from databases such as the Materials Project. Additionally, the tentative chemical reactions are not broken down into elementary steps. If the authors wish to support their mechanistic hypotheses with DFT, they should model the activation energies of elementary steps involved in the conversion of Li_2O into Li , and Li into Li_3N , and leverage these energetics into a micro-kinetic model.

Reviewer #3 (Remarks to the Author):

This manuscript presents a new route for ammonia production via laser-induced nitrogen fixation into metal oxides. The method using lithium oxide produced ammonia at a rate two orders of magnitude higher ($>40 \mu\text{mol s}^{-1} \text{cm}^{-2}$ with $A \sim 3.14 \text{ cm}^2$) than state-of-the-art technologies, including Haber-Bosch with much lower energy consumption. Unlike other technologies at the ambient conditions, such as electrochemical ammonia production, I see the merit of employing photons that easily break the triple bond of N_2 . The authors also conducted a systematic study to compare different metal oxides in ammonia production and identified an optimal medium. The comparison study also suggested a possible metal nitride formation mechanism supported by DFT calculation. This study introduces a new method for ammonia synthesis, which is quite different from other existing methods. I think that this work could be published in Nature Communications if the following points are carefully addressed.

Comments:

1. The authors mention that the new method could enable decentralized ammonia production. However, due to the nature of laser, it may be challenging to scale this method. Can authors provide information on how they envision the future of this technology and potential scalability challenges?
2. The manuscript mentioned that the Li_3N could be reused after dehydration of LiOH . However, comparing SEM images of Li_2O and Li_3N , the morphology changed dramatically. Have the authors tried to reuse the regenerated Li_2O for nitrogen fixation? How reversible is the process?
3. The authors suggested the bond dissociation energy is the critical factor for metal nitride formation. If Li_2O had the highest rate due to its low dissociation energy, why did ZnO not show the same trend? (Li-O: 341 kJ/mol, Zn-O: 284 kJ/mol)
4. Would there be any contribution of Ti film used for a support?
5. Why does ammonia production not linearly increase with the laser power density? (Fig. S3)
6. It seems laser scanning speed greatly contributes to the ammonia production and it changes depending on the laser power density or the metal oxide. For example, in Fig. S3, the slowest scanning speed (black, 0.17 mm s⁻¹) shows the least ammonia production at 68 and 152 W cm⁻² but the best or second to the best at 48, 91, 101, and 143 W cm⁻². Can you explain why it varies a lot?
7. According to Fig. S25, the ammonia concentration produced from MgO seems much higher than that from Li_2O . However, the performance is not even compatible with Li_2O shown in Fig. 4a. Why is that?
8. Is 5% photon utilization efficiency considered in the energy consumption calculated in Table S1? 332.7 kWh kg⁻¹ sounds very little. If it is not considered, it will be misleading.
9. As a person not familiar with this technology, it is hard to get a sense how long the process is. For example, how long does it take to scan the reaction area? These details will improve readers' understanding.

Response to the Reviewers' Comments

Many thanks to the reviewers for their valuable comments and suggestions. The followings are the point-by-point answers to their concerns. In the following the original comments are presented in black, our comments in blue and the actions taken in blue bold.

Response to Reviewer #1

Original comment:

This work by Wang et al. demonstrates ammonia production cycle by employing CO₂ laser with lithium oxide on Ti sheet. The high energy laser induces metal nitride from oxide in N₂ atmosphere, generating ammonia. Ammonia synthesis using a laser has been reported before and the yield rate is high enough. Thus, the idea in this work is a new challenge and creates another field for ammonia synthesis. The reviewer has no reservation for its publication after addressing following points.

Response: Thank you very much for your positive evaluation of our manuscript and constructive comments. We carefully revised the manuscript according to your advice and addressed comments point by point.

Comment 1:

Ammonia yield rate is high, but it appears to measure over a few microseconds. Is it possible for scale up? Although the laser has high energy, it seems to be unsuitable for scale up because laser is concentrated in a certain spot. Also, in terms of stability, all the chemicals are degraded if exposed to the laser for a long time.

Response: We conducted large-scale experiments under our current laboratory conditions and equipment. The figure below (Figure R1) shows that the production of ammonia increases proportionally with the extension of laser-induced acoustic time. This discovery has significant implications for future industrial applications and underscores the importance of continued research in this field.

Regarding the stability issue, in this study, lithium oxide reacts with nitrogen gas to produce lithium nitride **only** under laser induction. The generation of ammonia originates from the hydrolysis of lithium nitride. Furthermore, the detection of lithium nitride in the XRD and XPS analyses indicates that the generated lithium nitride is stable under laser radiation and not prone to decomposition.

According to your comment, we included Figure R1 as Figure S33 (Page 28) in the revised supporting information along with a related discussion in the revised manuscript (Page 5) as follows:

Paragraph 1, Page 5 in the revised manuscript: *“Furthermore, scaling up the LING process is both straightforward and feasible, with ammonia production of 1.3 mg achievable after only 78 seconds of irradiation (fig. S33). Given the ongoing optimization of laser technology and equipment, the LINF method shows great potential for practical application in industrial production.”*

Figure R1. Scale-up experiments: a) schematic diagram of prolonged scale-up experiments. b) The graph of the production of ammonia versus the irradiation time of the laser.

Comment 2:

If Li₂O acts as a catalyst, it must return to original chemical composition, whatever the chemical intermediates are formed. The authors proposed thermal dehydration, but did it consider when calculating energy consumption?

Response: Thank you for bringing this to our attention. Indeed, the thermal dehydration consumes additional energy, not considered in our previous calculations. To avoid any potential issues, we conducted the LINF process on hydrolyzed lithium hydroxide instead. Our results demonstrate that the production rate of laser-induced ammonia using lithium hydroxide is comparable to that using lithium oxide (Figure R2 and R3).

According to your comment, we included Figure R2 and R3 as Figure S31 and S32 (Page 27-28) in the revised supporting information along with a related discussion in the revised manuscript (Page 5) as follows:

Paragraph 1, Page 5 in the revised manuscript: *“Remarkably, in this lithium recycling approach, the lithium hydroxide obtained through the hydrolysis of lithium nitride can be stimulated by laser at room temperature, yielding ammonia similarly. Additionally, a considerably high yield of 40.3 $\mu\text{mol s}^{-1} \text{cm}^{-2}$ of ammonia was achieved under 200 °C (figs. S31-32).”*

Figure R2. a). Lithium cycling schematic diagram of the LING method; b). Ammonia yield rate by using lithium oxide and lithium hydroxide, where the lithium hydroxide is generated through the hydrolysis of lithium nitride. Ammonia yields of relatively high magnitude were observed at 200°C. In the laser-induced process, the presence of moisture, subsequent to lithium hydroxide dehydration, can influence the formation of lithium nitride. An increase in temperature can aid in the removal of moisture from the reaction interface.

Figure R3. XRD patterns of LiOH obtained via the hydrolysis process of lithium nitride and commercially available LiOH were analyzed and compared: The LiOH

sample obtained through the process of hydrolysis and subsequent drying of Li₃N contains a minor quantity of LiCO₃. In contrast, commercially available LiOH exhibits higher levels of lithium carbonate impurities. In addition, the lithium recovery rate achieved in the cycle process is an impressive 85%.

Comment 3:

There is a lot of information about yield rate, but no information about yield. The conversion of Li₂O is 5%, but how about the yield? Is the yield comparable to other synthesis methods? For readers' better understanding, comparison table with precedent works (methods, such as the thermochemical Haber-Bosch process, electrochemical process, mechanochemical process, and so on).

Response: In our work, the term "5%" refers to the photon utilization rate, with a corresponding Li₂O conversion rate of 11.2%. We appreciate the reviewers for raising concerns regarding the yield. XRD and XPS analysis confirmed a reaction selectivity of nearly 100% for lithium oxide converting into lithium nitride, with no significant by-products detected. Consequently, the yield is approximately 11.2%.

In terms of comparing with other methods, while there are various electrochemical methods that involve lithium mediation, the LINF method stands out as a unique approach (Table S1), with this study being the first to report on it. As such, information regarding yield from other reported methods has yet to be found.

According to your suggestion, we included the yield of the LINF method in the Table S1 of revised supporting information.

Response to Reviewer #2

Original comment:

In manuscript NCOMMS-23-08213, Wang and coworkers describe an alternative cyclic ammonia synthesis strategy that uses focused light from a laser to convert Li_2O into a reactive lithium surface that can dissociate the N_2 bond and form a Li_3N intermediate. It is well known that Li_3N can then be easily hydrolyzed to NH_3 and LiOH , and Li_2O can be regenerated via subsequent thermal dehydration. The approach described here is reminiscent of other lithium cycling strategies such as that reported by McEnaney, et al (ref 17), with the primary new idea being the use of laser light rather than electricity to reduce oxidized forms of lithium into lithium nitride.

This work is interesting and potentially suitable for this venue, but I have a couple of concerns that must be addressed before I can endorse publication:

Response: Many thanks for your careful review of our manuscript and for giving constructive comments. We have addressed your comments point by point and carefully revised the manuscript according to your advice.

Comment 1:

In Figure 3c, the authors show that one of the main benefits of their laser-induced nitrogen fixation approach is the high rate of ammonia production compared to various alternatives including the lithium-mediated electrocatalytic approach (e.g. refs 9-11). While this is true for most of the electrocatalytic strategies in the literature, in my view authors should actually be comparing their work against the cycling strategy reported by McEnaney, et al (ref 17), which in my view is the closest technology and would be the direct competitor of the one proposed here. In other words, how does the rate of $\text{Li}_2\text{O} \rightarrow \text{Li (s)} \rightarrow \text{Li}_3\text{N}$ via laser-induced activation compare to the rate of lithium ions electroplating and reacting with N_2 at -3V ($\text{Li}^+ \rightarrow \text{Li (s)} \rightarrow \text{Li}_3\text{N}$)? The subsequent hydrolysis of Li_3N will be the same (fast) in both approaches.

Response: We express our gratitude to you for acknowledging the significance of the referenced literature, which provides invaluable insights. It is noteworthy that the lithium cycle approach, although innovative, produces a comparatively lower quantity of ammonia. The disparity arises due to the time-consuming nature of two-step process involved in their method: the molten salt electrolysis and lithium nitride synthesis (LiOH-Li-Li₃N). Conversely, in our laser-induced process, the dissociation of lithium oxide into lithium nitride (Li₂O-Li-Li₃N) is accomplished almost instantaneously, completing the entire experimental process in one step. According to your suggestion, a comprehensive comparison of the rates is presented in Table S1:

Comment 2:

Since the primary novelty of this work is the laser-induced activation of Li₂O, I believe that there needs to be a more thorough discussion of the techno-economic benefits of using a laser compared to electricity/electrochemistry. The authors suggest that this technology will be economically relevant, but this statement needs more support.

Response: To underscore economic relevance of this technology, we conducted a thorough comparison with the current market price of ammonia, as depicted in Figure R4. Currently, our method's cost per ton of ammonia production is higher than traditional methods due to its relatively low photon utilization rate, only 5%. However, if the photon utilization rate reaches 40%, it can be lower than industrial production costs. This significant finding not only demonstrates the potential economic benefits but also emphasizes the need for further research and development of our method, as it holds tremendous application potential.

According to your suggestion, we have revised the manuscript with the following text (Page 5) and included Figure R4 as Figure S35 (Page 29) in the revised supporting information:

Paragraph 1, Page 5 in the revised manuscript: ***“In terms of the economic benefits, achieving a photon utilization rate of 40% or higher can yield cost***

advantages surpassing current industrial ammonia production in the market.”

Figure. R4. Ammonia production cost analysis was conducted for the LINF process, considering different photon utilization efficiencies. A comparison was made between the resulting ammonia costs in the LINF process and conventional industrial methods: In the article, we discussed the potential application of future solar-pumped laser converters. Therefore, the price per kilowatt-hour (kWh) mentioned here is derived from the current cost of solar photovoltaic power generation, which amounts to \$0.031 per kWh (the price comes from Wikipedia: https://en.wikipedia.org/wiki/Cost_of_electricity_by_source). The average price of ammonia in 2022 is \$1400 per ton. (the price comes from “Farmdoc Daily”: <https://farmdocdaily.illinois.edu/2022/09/fertilizer-prices-rates-and-costs-for-2023.html>)

Comment 3:

The proposed mechanism and associated DFT modeling in Figure 4 are very limited and do not add much beyond what is already known. The relative thermodynamics of Li₂O, Li₃N, and Li can easily be calculated from databases such as the Materials Project. Additionally, the tentative chemical reactions are not broken down into elementary steps. If the authors wish to support their mechanistic hypotheses with DFT, they should model the activation energies of elementary steps involved in the conversion of Li₂O into Li, and Li into Li₃N, and leverage these energetics into a micro-kinetic model.

Response: We agree with your comment that more details on the mechanism are needed to further understand the process beyond the bulk values that can be found in databases. However, we believe that the path suggested by the reviewer consisting of the DFT evaluation of the full reaction path through conventional approaches including coupling to microkinetics is not well-suited for this case. Microkinetic modelling is typically effective for addressing thermal processes where the number of active sites remain constant. In the present case, the activation takes place via a laser with IR energy. This implies that the mechanism is rather related to changes in the catalyst's structure induced by phonon excitations. This introduces a new alternative path with rules that are different from the standard thermal processes. Given the novelty of the mechanism, we have focused on identifying the most likely ways how the material is switched on and can activate N₂, which is the highest kinetic barrier during the process (Science 307, 555–558 (2005)).

Therefore, we expanded our Density Functional Theory (DFT) simulations to investigate the activation of Li₂O. Specifically, we ran simulations on the (111) and (211) facets of Li₂O and analysed the phonon density of states to identify the vibrational modes that correspond to the laser energy. The analysis of the vibrational normal modes suggests that there are at least three plausible defects—Li adatom, O peroxide, and anti-site Li-O exchange—that fall in the CO₂ infrared laser frequency ranges. All three defects can activate N₂, resulting in the elongation of bond distances from 1.115 Å in the gas phase to 1.165 Å for the adatom, and further to 1.280 Å for the other two defects. These extended N-N bond distances serve as the starting point for the observed reactivity. This initiation step, differs completely from the traditional thermal pathways, plays a crucial role in enhanced understanding of the laser-induced catalytic process.

We have incorporated the DFT simulations into the manuscript by making the following changes: Figure 4 was modified to illustrate the activation modes for the step surface (Figure 4c) and the corresponding structures for the Li adatom defect (Figure 4d). Additional text describing the DFT simulations have been added to the

manuscript (Page 4) along with figures S26-S29 and tables S4-S13 in the Supporting Information. Finally, all the DFT simulations are uploaded to open source iochem-BD database which can be accessed through the following link: <https://iochem-bd.iciq.es/browse/review-collection/100/61365/1612d1ba98d4b331c88b3aa9>.

Page 11 in the revised manuscript:

Fig. 4. Proposed mechanism of laser-induced nitrogen fixation. *a.* Ammonia yield rate by using lithium oxide, magnesium oxide, aluminum oxide, calcium oxide and zinc oxide as precursor mediators: a scanning speed of 0.17 mm s⁻¹ and an N₂ pressure at 7.5 bar; *b.* The number of photons gathered on a single Li₂O unit cell (purple ball represents lithium atom; red ball represents oxygen atom) by using laser power of 118 kW cm⁻² with a pulse (75 μs) and a laser focus diameter of 170 μm; *c.* The phonon density of states of Li₂O (211) surface, the inset of the figure shows the atom movements corresponding to high frequency mode (843.7 cm⁻¹); *d.* Nudged elastic band calculation for the generation of the adatom (Li_{ad}) defect in the Li₂O (211) surface. N₂ adsorption in the vacancy close to the adatom with N-N bond distance stretching to 1.165 Å (gas phase N-N = 1.115 Å). Li adatom is shown in the dark purple color, lithium atoms as purple, oxygen atoms as red, nitrogen atoms as blue.

e. Schematic illustration of multiphoton absorption by Li_2O as an example during the laser-induced process. After each oxygen-lithium bond absorbs at least tens of photons (the energy of each photon is 0.12 eV), the oxygen-lithium bond is dissociated, and the excited state lithium transitions to a lower energy level and emits bright light. Part of the excited state lithium is combined with nitrogen gas forms lithium nitride and emits visible red light.

Response to Reviewer #3

Original comment:

This manuscript presents a new route for ammonia production via laser-induced nitrogen fixation into metal oxides. The method using lithium oxide produced ammonia at a rate two orders of magnitude higher ($>40 \mu\text{mol s}^{-1} \text{cm}^{-2}$ with $A \sim 3.14 \text{cm}^2$) than state-of-the-art technologies, including Haber-Bosch with much lower energy consumption. Unlike other technologies at the ambient conditions, such as electrochemical ammonia production, I see the merit of employing photons that easily the triple bond of N_2 . The authors also conducted a systematic study to compare different metal oxides in ammonia production and identified an optimal medium. The comparison study also suggested a possible metal nitride formation mechanism supported by DFT calculation. This study introduces a new method for ammonia synthesis, which is quite a different from other existing methods. I think that this work could be published in Nature Communications if the following points are carefully addressed.

Response: Thank you for your positive evaluation and very helpful suggestions. We have carefully revised the manuscript according to your advice and addressed your comments point by point as follows:

Comment 1:

The authors mention that the new method could enable decentralized ammonia production. However, due to the nature of laser, it may be challenging to scale this method. Can authors provide information on how they envision the future of this technology and potential scalability challenges?

Response: We conducted large-scale experiments under our current laboratory conditions and equipment. The figure below (Figure R1) shows that the production of ammonia increases proportionally with the extension of laser-induced acoustic time. This discovery has significant implications for future industrial applications and

underscores the importance of continued research in this field.

According to your comment, we included Figure R1 as Figure S33 (Page 28) in the revised supporting information along with a related discussion in the revised manuscript (Page 5) as follows:

Paragraph 1, Page 5 in the revised manuscript: *“Furthermore, scaling up the LING process is both straightforward and feasible, with ammonia production of 1.3 mg achievable after only 78 seconds of irradiation (fig. S33). Given the ongoing optimization of laser technology and equipment, the LINF method shows great potential for practical application in industrial production.”*

Figure R1. Scale-up experiments: a) schematic diagram of prolonged scale-up experiments. b) The graph of the production of ammonia versus the irradiation time of the laser.

Comment 2:

The manuscript mentioned that the Li_3N could be reused after dehydration of LiOH . However, comparing SEM images of Li_2O and Li_3N , the morphology changed dramatically. Have the authors tried to reuse the regenerated Li_2O for nitrogen

fixation? How reversible is the process?

Response: We conducted an additional supplementary experiment to complete the lithium cycle. However, taking into account the additional heat energy consumption resulting from the pyrolysis of LiOH to Li₂O, we opted to directly utilize LiOH following the hydrolysis of lithium nitride. Through repeating the LINF process, we obtained comparable yields of ammonia. As a result, we accomplished the Li₂O → Li₃N ⇌ LiOH cycle while circumventing the need for extra energy consumption associated to pyrolysis.

According to your comment, we included Figure R2 and R3 as Figure S31 and S32 (Page 27-28) in the revised supporting information along with a related discussion in the revised manuscript (Page 5) as follows:

Paragraph 1, Page 5 in the revised manuscript: “*Remarkably, in this lithium recycling approach, the lithium hydroxide obtained through the hydrolysis of lithium nitride can be stimulated by laser at room temperature, yielding ammonia similarly. Additionally, a considerably high yield of 40.3 μmol s⁻¹ cm⁻² of ammonia was achieved under 200 °C (figs. S31-32).*”

Figure R2. Ammonia yield rate by using lithium oxide and lithium hydroxide, where the lithium hydroxide is generated through the hydrolysis of lithium nitride. Ammonia yields of relatively high magnitude were observed at 200°C. In the laser-induced process, the presence of moisture, subsequent to lithium hydroxide dehydration, can influence the formation of lithium nitride. An increase in temperature can aid in the removal of moisture from the reaction interface.

Figure R3. XRD patterns of LiOH obtained via the hydrolysis process of lithium nitride and commercially available LiOH were analyzed and compared: The LiOH sample obtained through the process of hydrolysis and subsequent drying of Li₃N contains a minor quantity of LiCO₃. In contrast, commercially available LiOH exhibits higher levels of lithium carbonate impurities.

Comment 3:

The authors suggested the bond dissociation energy is the critical factor for metal nitride formation. If Li₂O had the highest rate due to its low dissociation energy, why did ZnO not show the same trend? (Li-O: 341 kJ/mol, Zn-O: 284 kJ/mol).

Response: I appreciate you for bringing up this point. Undoubtedly, the breaking of

metal-oxygen bonds plays a crucial role in the LINF process. However, it is essential to note that the reactivity of zero-valent metals with nitrogen is another significant factor influencing nitride formation. Among these metals, lithium exhibits the highest level of reactivity and can even react with nitrogen at room temperature, leading to the formation of lithium nitride (ref.17, *Energy Environ. Sci.* 10, 1621–1630 (2017)). In contrast, the formation of zinc nitride necessitates the reaction of zinc with ammonia at temperatures exceeding 600 °C (*J. Cryst. Growth* **312**, 1838–1843 (2010)).

According to your comment and to make a clear discussion on this aspect, the supplementary discussions were added in revised manuscript (**Pages 3**) and in the revised supporting information (**Pages 31**), as following:

Paragraph 3, Page 3 in the revised manuscript: *“Although zero-valent lithium is not detected, zero-valent Mg, Al, Ca, and Zn are left to be detected when using other metal oxides (MgO, Al₂O₃, CaO, and ZnO) due to their lower reactivity towards N₂ compared to lithium metal (figs. S11-24, supplementary note 1), which agree well with the fact that using Li₂O as a medium produces NH₃ with the highest rate (Fig. 4a, fig. S25).”*

The caption of **Table S2 (Page 31)** in the revised supporting information: *“It is essential to note that besides the breaking of metal-oxygen bonds plays a crucial role in the LINF process, the reactivity of zero-valent metals with nitrogen is another significant factor influencing nitride formation. Among these metals, lithium exhibits the highest level of reactivity and can even react with nitrogen at room temperature, leading to the formation of lithium nitride²⁸. In contrast, the formation of zinc nitride necessitates the reaction of zinc with ammonia at temperatures exceeding 600 °C²⁹.”*

Comment 4:

Would there be any contribution of Ti film used for a support?.

Response: The choice of titanium sheets as a substrate is primarily attributed to their excellent stability and high melting point. Furthermore, the titanium nitride produced as a result of the LINF process exhibits enhanced stability, thereby augmenting the substrate's overall stability. This desirable characteristic allows for the repeated utilization of the substrate, ensuring its prolonged functionality.

According to your comment, additional discussion has added in the method section of revised supporting information (**Page 3**) as following:

Page 3 in the revised supporting information: *“The choice of titanium sheets as a substrate is primarily attributed to their excellent stability and high melting point. Furthermore, the titanium nitride produced as a result of the LINF process exhibits enhanced stability, thereby augmenting the substrate's overall stability. This desirable characteristic allows for the repeated utilization of the substrate, ensuring its prolonged functionality.”*

Comment 5:

Why does ammonia production not linearly increase with the laser power density? (Fig. S3)

Response: The primary role of laser irradiation is the generation of defects in the form of zero-valent metals which are able to react with N₂, see new DFT simulations. The formation of nitrides, and subsequently the production of ammonia, is contingent upon both the reactivity of the metal and the reaction time (in this case, determined by the laser scanning speed). It is important to note that the dissociation of the metal-oxygen bonds necessitates a critical laser power density. Once all the oxides have been induced and excited within the laser's range, further increasing the laser power density has minimal impact on promoting subsequent nitride formation.

According to your reminder, supplemented discussion was added in the revised supporting information (Page 7) as follows:

The caption of Figure S3 (Page 7) in the revised supporting information: *“The*

primary role of laser irradiation is to facilitate the dissociation of metal-oxygen bonds, leading to the generation of zero-valent metals. The formation of nitrides, and subsequently the production of ammonia, is contingent upon both the reactivity of the metal and the reaction time (in this case, determined by the laser scanning speed). It is important to note that the dissociation of the metal-oxygen bonds necessitates a critical laser power density. Once all the oxides have been induced and excited within the laser's range, further increasing the laser power density has minimal impact on promoting subsequent nitride formation.”

Comment 6:

It seems laser scanning speed greatly contributes to the ammonia production and it changes depending on the laser power density or the metal oxide. For example, in Fig. S3, the slowest scanning speed (black, 0.17 mm s^{-1}) shows the least ammonia production at 68 and 152 W cm^{-2} but the best or second to the best at 48, 91, 101, and 143 W cm^{-2} . Can you explain why it varies a lot?

Response: We thank you very much for your careful review and invaluable comment. As discussed in the above question, the production yield of ammonia is influenced by several factors. These include the laser power density, which determines the energy required to break the metal-oxygen bond, the scanning speed of the laser, which affects the overall process time including metal oxide dissociation, and the reaction time of the activated zero-valent metal with nitrogen. Furthermore, the reactivity of the metal plays a crucial role in ammonia production.

In essence, there exists an optimal scanning speed for a specific power density. Lower power densities necessitate longer reaction times, resulting in the higher ammonia yields. Conversely, higher power densities can achieve satisfactory results with shorter irradiation times, thereby consuming less energy and enhancing productivity. Therefore, multiple experimental parameters influencing each other lead to the results in Fig. S3: the slowest scanning speed (black, 0.17 mm s^{-1}) shows the least ammonia production at 68 and 152 W cm^{-2} but the best or second to the best at

48, 91, 101, and 143 W cm⁻².

According to your comment, a supplemented discussion was added in the revised supporting information (Page 8) as follows:

The caption of Figure S3 (Page 8) in the revised supporting information: *“In addition, the production yield of ammonia is influenced by several factors. These include the laser power density, which determines the energy required to break the metal-oxygen bond, the scanning speed of the laser, which affects the overall process time including metal oxide dissociation, and the reaction time of the activated zero-valent metal with nitrogen. Furthermore, the reactivity of the metal plays a crucial role in ammonia production. In essence, there exists an optimal scanning speed for a specific power density. Lower power densities necessitate longer reaction times, resulting in the higher ammonia yields. Conversely, higher power densities can achieve satisfactory results with shorter irradiation times, thereby consuming less energy and enhancing productivity. Therefore, multiple experimental parameters influencing each other leads to the results in Fig. S3: the slowest scanning speed (black, 0.17 mm s⁻¹) shows the least ammonia production at 68 and 152 W cm⁻² but the best or second to the best at 48, 91, 101, and 143 W cm⁻².”*

Comment 7:

According to Fig. S25, the ammonia concentration produced from MgO seems much higher than that from Li₂O. However, the performance is not even compatible with Li₂O shown in Fig. 4a. Why is that?

Response: We are sorry for not stating this part clearly. Different volumes of water were used to hydrolyze lithium nitride and other metal nitrides, thus leading to a misleading figure.

Following your comment, supplemented descriptions were noted in the method section (Page 3) and Figure S25's caption (Page 23) of revised supporting information

as follows:

Page 3 in the revised supporting information: “*The generated ammonia was detected and quantified after the hydrolysis of Li₃N in 100 mL of 0.05 M hydrochloric acid solution. The generated ammonia from other metal nitrides was detected and quantified after the hydrolysis of metal nitrides in 15 mL of 0.05 M hydrochloric acid solution.*”

The caption of Figure S25 (Page 23) in the revised supporting information: “*It is worth noting that the generated ammonia was detected and quantified after the hydrolysis of Li₃N in 100 mL of 0.05 M hydrochloric acid solution. The generated ammonia from other metal nitrides was detected and quantified after the hydrolysis of metal nitrides in 15 mL of 0.05 M hydrochloric acid solution.*”

Comment 8:

Is 5% photon utilization efficiency considered in the energy consumption calculated in Table S1? 332.7 kWh kg⁻¹ sounds very little. If it is not considered, it will be misleading.

Response: We calculated the energy consumption based on real laser radiation power. Therefore, 5% photon utilization efficiency considered in the energy consumption calculated in Table S1.

Following your comment, a supplemented description was noted in the method section of revised supporting information (Page 4) as follows:

Page 4 in the revised supporting information: “*The correlated energy consumption is given by*

$$E = \frac{pt}{cVM}$$

where p is the laser power used, t is the time of the laser-induced process, c is the measured NH₃ concentration, V is the volume of the hydrochloric acid solution and M is the relative molecular mass of NH₃.”

Comment 9:

As a person not familiar with this technology, it is hard to get a sense how long the process is. For example, how long does it take to scan the reaction area? These details will improve readers' understanding.

Response: We are sorry for not stating this part clearly. The reaction time is determined by laser sweep, details added to Table S14.

According to your comment, a supplemented description was noted in the method section of revised supporting information (Page 3) as follows:

Page 3 in the revised supporting information: “*The setting laser parameters: laser power between 10.8 – 37.3 W and the laser scanning speed between 0.17 – 1.36 mm s⁻¹ (corresponding reaction times of 53 – 7 seconds to scan the reaction area in this work (Table S14)), adjusted for the experiments to improve the ammonia yield rate.*”

REVIEWERS' COMMENTS

Reviewer #1 (Remarks to the Author):

The authors well responded to the reviewers' concerns, and thus it is now ready for publication.

Reviewer #2 (Remarks to the Author):

The authors have addressed my comments and I now support publication of the manuscript as submitted.

Reviewer #3 (Remarks to the Author):

The authors addressed all my concerns. This is a great piece of work. Congratulations to the team.